# A retrospective observational study of enhanced recovery after surgery in older patients undergoing elective colorectal surgery

Katleen Fagard[1]*, Albert Wolthuis[2], Marleen Verhaegen[3], Johan Flamaing[1,4], Mieke Deschodt[4,5]

1 Department of Geriatric Medicine, University Hospitals Leuven, Leuven, Belgium, 2 Department of Abdominal Surgery, University Hospitals Leuven, Leuven, Belgium, 3 Department of Anesthesia, University Hospitals Leuven, Leuven, Belgium, 4 Division of Gerontology and Geriatrics, Department of Public Health & Primary Care, KU Leuven, Leuven, Belgium, 5 Department of Public Health, Institute of Nursing Science, University of Basel, Basel, Switzerland

* katleen.fagard@uzleuven.be

**Data Availability Statement:** All relevant data are within the paper and supporting information files.

**Funding:** The author(s) received no specific funding for this work.

## Abstract

### Background

Enhanced recovery programs (ERPs) in colorectal surgery have demonstrated beneficial effects on postoperative complications, return of bowel function, length of stay, and costs, without increasing readmissions or mortality. However, ERPs were not specifically designed for older patients and feasibility in older patients has been questioned.

### Aim

The aim of this study was to assess ERP adherence and outcomes in older patients and to identify risk factors for postoperative complications and prolonged length of stay.

### Method

Retrospective analysis of consecutive patients (≥70 years) undergoing elective colorectal resection in a tertiary referral hospital in 2017.

### Results

Ninety-six patients were included. Adherence rates were above 80% in 18 of 21 ERP interventions considered. The lowest adherence rates were noted for preoperative carbohydrate loading and cessation of intravenous fluids. Postoperative complications (Clavien-Dindo ≥2) and prolonged postoperative length of stay (>75th percentile) were observed in 39.6% and 26.3%, respectively. Median length of stay was 7 days. The 30-day mortality, readmission and reoperation rates were 2.1%, 12.6% and 8.3%, respectively. Multivariable analysis indicated that polypharmacy and site of surgery were independent risk factors for postoperative complications, while higher age, American Society of Anesthesiologists class and

**Competing interests:** The authors have declared that no competing interests exist.

preoperative radiotherapy were independent risk factors for prolonged postoperative length of stay.

## Conclusion

ERP adherence in older patients undergoing colorectal resection is high and ERP is therefore considered feasible. Postoperative complications and prolonged postoperative length of stay are common, so at risk patients should be targeted with tailored geriatric interventions.

## Introduction

Population growth, demographic ageing and advances in surgical and anesthetic techniques have caused a marked increase in the demand for surgical procedures in older persons [1–3]. As ageing is associated with a decrease in physiologic reserve and higher rates of comorbidity, many older patients are more susceptible to adverse postoperative outcomes, such as medical and surgical complications, prolonged hospital stay, loss of independence in activities of daily living, and need for institutionalization. This presents organizational and socio-economic challenges to our health care system [3–6].

Surgical teams have developed and implemented multidisciplinary care programs, known as Fast Track Surgery, Enhanced Recovery Programs (ERPs) or Enhanced Recovery After Surgery (ERAS®) programs, resulting in improved patient outcomes and reduced lengths of stay (LOS) [7, 8]. ERPs include evidence-based changes in traditional care, such as pre-admission counselling, avoidance of mechanical bowel preparation, shortened fasting, a carbohydrate drink two hours before surgery, avoidance of fluid overload, minimal invasive surgery, avoidance or early removal of drains and catheters, opioid-sparing multimodal analgesia, early feeding and mobilization. A recent meta-analysis has demonstrated that ERPs in colorectal surgery are associated with a significant reduction in LOS, postoperative complications, total cost of hospital stay, as well as earlier return of gastro-intestinal function, without increasing 30-day readmission or mortality rates [9]. However, ERPs were not specifically designed for frail older patients and data from these programs in older patients are scarce [10–12]. It might be necessary to adapt ERPs for older patients with multi-morbidity, functional, cognitive or psychosocial problems.

The objectives of the present study were to assess ERP adherence and outcomes in older patients undergoing colorectal resection, and to identify risk factors for postoperative complications and prolonged LOS.

## Materials and methods

### Study design and setting

Single-center retrospective observational cohort study. The study was conducted in the University Hospitals Leuven. Patients were selected from a database with planned operations in 2017. Data were collected from the patients' electronic medical records. The medical ethics committee of the University Hospitals Leuven approved the study (S61709).

### Sample

Patients aged 70 years and over who underwent elective colorectal resections in an ERP were eligible for inclusion. Patients undergoing emergency surgery, hyperthermic intraperitoneal

chemotherapy, adhesiolysis, stoma closure or transit repair, rectopexy or prolapse surgery, transanal procedures, proctological operations or local stoma procedures were excluded.

## Enhanced recovery program

The standard ERP for colorectal surgical patients at the time of the study is detailed in S1 Table. The program was applied to all adult patients, irrespective of age. ERP components are divided in 20 intervention categories, based on the ERAS® guidelines (2012) that were valid at that time [13, 14].

## Variables and measurements

**Demographic variables.** The following demographic data were collected: age, gender, and living situation: at home, assisted living facility, nursing home.

**Clinical baseline variables.** Preoperative baseline characteristics retrieved from the electronic medical records were: the Flemish version of the Triage Risk Screening Tool (fTRST) [15], number of medications, nutritional risk score (NRS-2002) initial screening items [16], Charlson Comorbidity Index (CCI) [17], age adjusted CCI (ACCI) [18], height, weight, cancer or benign diagnosis, and preoperative treatment for cancer with chemo- or radiotherapy.

The fTRST screens older patients for their risk of hospitalization associated functional decline, recurrent hospital admissions or nursing home admission by scoring the following risk factors: living alone, cognitive impairment, impaired mobility, polypharmacy ($\geq$5 medications) and hospitalization within 3 months. Patients scoring $\geq$2 on the fTRST are at increased risk [15]. The NRS-2002 was developed to detect undernutrition in the hospital setting and includes the following items: Body Mass Index (BMI) <20.5, weight loss within the last 3 months, reduced dietary intake in the last week, severely ill. A patient scoring positive on at least one these risk factors in combination with planned major abdominal surgery is undernourished or at risk of undernutrition [16]. Both are routinely scored by the nursing team on admission to our hospital. The CCI as well as the ACCI were calculated based upon the comorbidities encountered in the medical records. The ACCI adds 3 points to the CCI for age 70–79, and 4 points for age $\geq$80 [18]. Height and weight were used to calculate the BMI. BMI results were classified using the World Health Organization criteria: underweight <18.5 kg/m$^2$, adequate 18.5–24.9 kg/m$^2$, overweight 25–29.9 kg/m$^2$ or obese $\geq$30 kg/m$^2$ [19].

**Surgery-related variables.** The following surgery-related variables were collected: American Society of Anesthesiologists (ASA) class [20], type of surgery (ileocecal resection or right colectomy or left colectomy, sigmoid resection or Hartmann, rectal surgery, total or proctocolectomy), site of surgery (colon, rectum), surgical approach (laparoscopic, open or converted), stoma creation, other surgical procedures (resection of structures other than those mentioned in 'type of surgery').

**Adherence.** Adherence to 21 ERP interventions was derived from the electronic medical records and reported as the percentage of patients that adhered to the intervention. The interventions considered were: 1) Preadmission education by ERP nurse, 2) No mechanical bowel preparation in colonic surgery, 3) Carbohydrate loading 3 hours prior to surgery, 4) No sedative or anxiolytic premedication, 5) Thromboprophylaxis with low molecular weight heparin, 6) Antimicrobial prophylaxis, 7) Postoperative nausea and vomiting (PONV) prophylaxis administered in patients with $\geq$1 risk factor on the Apfel Score [21, 22], 8) Planned laparoscopic operation, 9) No nasogastric tube after reversal of anesthesia, 10) Prevention of intraoperative hypothermia, 11) Cessation of intravenous fluids by postoperative day (POD) 3, provided removal of PCEA or patient controlled intravenous analgesia (PCIA), 12) No abdominal drain in colonic surgery, 13) Removal of urinary catheter on POD 1 in laparoscopic

colonic surgery and on POD 3 in open colonic surgery, provided PCEA removal, 14) Patient controlled epidural analgesia (PCEA) activated before first incision in open surgery, provided patient without long-term anticoagulant therapy, 15) Oral food on POD 1, 16) Oral food on POD 2, 17) Oral food on POD 3, 18) Glucose day profile in patients with diabetes, 19) Out of bed on POD 1, 20) Out of bed on POD 2, 21) Out of bed on POD 3.

**Outcome variables.**   The primary endpoints of this study were the occurrence of Clavien-Dindo grade 2 and above in-hospital postoperative complications and prolonged postoperative LOS, i.e. LOS exceeding the 75th percentile.

The secondary outcome variables included: In-hospital complications that occurred during or after surgery and their severity grading according to the Clavien-Dindo classification [23, 24], the Comprehensive Complication Index [25], mortality within the first 30 postoperative days, postoperative LOS, 30-day readmission rate, unplanned reoperation within the first 30 postoperative days. Postoperative LOS was defined as the number of postoperative days spent in the hospital until discharge or until transfer to a rehabilitation unit. The 30-day readmission rate was defined as the number of patients with unplanned readmissions to the hospital within 30 days of discharge due to a complication of the primary operation.

## Data analysis

Continuous variables were reported as medians with interquartile ranges (IQR). Categorical variables were reported as numbers and percentages. The demographic, baseline and surgery-related variables were studied as predictors for the primary outcomes in univariable analysis. Dichotomous variables were compared using Chi-squared or Fisher's exact tests. Nominal variables were compared using Chi-squared tests. Ordinal and non-normally distributed continuous variables were compared using Mann-Whitney U tests. To determine independent predictors for our primary outcomes, all variables were entered in a multivariable forward logistic regression model. P-values, Odds Ratios (OR), and 95% confidence intervals (CI) are reported. All tests were 2-tailed, assuming a 5% significance level. All analyses were performed using SPSS version 20.0 (SPSS Inc., Chicago, IL).

## Results

### Description of the sample

Ninety-six patients were included in the study (Fig 1). Their median age was 77 (IQR 73–82) years, and 50% were female (Table 1). The majority (92.7%) lived at home before admission. fTRST scoring was positive in 49% of the patients: 26% was living alone, 4.2% had cognitive impairment, 24% had impaired mobility, 62.5% took 5 or more medications, and 40.6% had been hospitalized in the last 3 months. The NRS-2002 identified 33 patients (34.4%) with undernutrition or risk for undernutrition. Median BMI was 25.3 kg/m$^2$ (IQR 23.0–28.7). Fifty-eight patients (60.4%) had at least two chronic diseases in the CCI list and 31 patients (32.3%) scored above the median CCI score of 3. Forty-seven patients (49.0%) had ACCI scores above the median of 6. A majority of patients (68.7%) were classified in ASA class 3 or 4. One quarter of the patients underwent surgery for benign disease; the remaining patients had colorectal cancer. The majority of operations were planned laparoscopically: 82.0% of colonic operations (50/61) and 88.6% (31/35) of rectal operations. In 10.0% of colonic operations (5/50) and 16.1% of rectal operations (5/31) the laparoscopic procedure was converted to open surgery. The reasons for conversion were obesity (n = 2), extensive adhesions (n = 5), extensive malignancy (n = 2), and difficult splenic flexure mobilisation (n = 1).

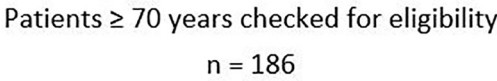
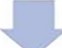
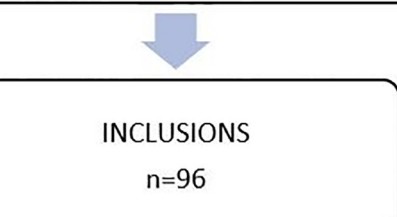

Patients ≥ 70 years checked for eligibility
n = 186

EXCLUSIONS (n = 90)
- emergency surgery (n = 1)
- HIPEC (n = 4)
- adhesiolysis (n = 1)
- stoma closure, transit repair (n = 30)
- rectopexy, prolaps surgery (n = 11)
- TAE, TAMIS, anal surgery or local procedure stoma (n = 18)
- not hospitalised in the abdominal surgery department (n = 6)
- non-colorectal surgery or surgery omitted (n = 8)
- double in list or reoperation for surgical complication (n = 11)

INCLUSIONS
n=96

**Fig 1. Flowchart of the study recruitment.** HIPEC: hyperthermic intraperitoneal chemotherapy; TAE: transanal excision; TAMIS: transanal minimally invasive surgery.

## Adherence to the ERP

Table 2 summarizes 21 ERP interventions with their level of adherence. The majority of interventions had high adherence rates ranging between 80.0 and 100%. Lower adherence rates were noted for urinary catheter removal (77.5%), for carbohydrate loading (69.8%), and for cessation of intravenous fluids by POD 3 (54.7%). There was no significant difference in carbohydrate loading in patients with or without diabetes: 65.0% of patients with diabetes versus 71.1% of patients without diabetes (p = 0.595). In the subgroup of patients without complications needing medical or surgical treatment (Clavien-Dindo <2) the adherence rate for cessation of intravenous fluids was 63.8% and the protocol for urinary catheter removal was applied in 87.7%. The median POD of urinary catheter removal was 1 (IQR 1–3) in patients with Clavien-Dindo <2 postoperative complications and 3 (IQR 1–6.75) in patients with Clavien-

**Table 1. Baseline characteristics of the total study sample and in relation to the primary outcomes (univariable analysis).**

| | All patients (n = 96) | CD < 2 POCs (n = 58) | CD ≥ 2 POCs (n = 38) | P-value | LOS ≤ P75[1] (n = 70) | LOS > P75[1] (n = 25) | P-value |
|---|---|---|---|---|---|---|---|
| Age, median (IQR) | 77 (73–82) | 76 (72–81) | 80.5 (73.8–84) | **0.035** | 76 (72–81) | 80 (75–84) | **0.011** |
| Gender, n (%) | | | | | | | |
| Male | 48 (50.0) | 29 (50.0) | 19 (50.0) | 1.000 | 34 (48.6) | 13 (52.0) | 0.769 |
| Female | 48 (50.0) | 29 (50.0) | 19 (50.0) | | 36 (51.4) | 12 (48.0) | |
| Living situation, n (%) | | | | | | | |
| at home | 89 (92.7) | 55 (94.8) | 34 (89.5) | 0.556 | 66 (94.3) | 22 (88.0) | 0.272 |
| assisted living facility | 3 (3.1) | 1 (1.7) | 2 (5.3) | | 1 (1.4) | 2 (8.0) | |
| nursing home | 4 (4.2) | 2 (3.4) | 2 (5.3) | | 3 (4.3) | 1 (4.0) | |
| fTRST-score ≥ 2, n (%) | 47 (49.0) | 24 (41.4) | 23 (60.5) | 0.066 | 28 (40.0) | 18 (72.0) | **0.006** |
| fTRST: living alone, n (%) | 25 (26.0) | 13 (22.4) | 12 (31.6) | 0.317 | 17 (24.3) | 7 (28.0) | 0.714 |
| fTRST: cognitive impairment, n (%) | 4 (4.2) | 1 (1.7) | 3 (7.9) | 0.297 | 3 (4.3) | 1 (4.0) | 1.000 |
| fTRST: impaired mobility, n (%) | 23 (24.0) | 12 (20.7) | 11 (28.9) | 0.654 | 11 (15.7) | 11 (44.0) | **0.004** |
| fTRST: polypharmacy[2], n (%) | 60 (62.5) | 31 (53.4) | 29 (76.3) | **0.024** | 40 (57.1) | 20 (80.0) | **0.042** |
| fTRST: recent hospitalisation[3], n (%) | 39 (40.6) | 23 (39.7) | 16 (42.1) | 0.811 | 26 (37.1) | 12 (48.0) | 0.342 |
| Number of medications, median (IQR) | 6 (4–8) | 5 (3–8) | 7 (4.75–8) | **0.040** | 5 (3–8) | 6 (5–8) | 0.243 |
| Nutrition risk[4], n (% | | | | | | | |
| 1 screening item positive | 27 (28.1) | 14 (24.1) | 13 (34.2) | 0.472 | 17 (24.3) | 9 (36.0) | 0.538 |
| 2 screening items positive | 5 (5.2) | 3 (5.2) | 2 (5.3) | | 4 (5.7) | 1 (4.0) | |
| 3 screening items positive | 1 (1.0) | 1 (1.7) | 0 (0.0) | | 1 (1.4) | 0 (0) | |
| CCI, median (IQR) | 3 (2–4) | 3 (2–4) | 3 (2–5) | 0.077 | 3 (2–4) | 3 (2–4) | 0.378 |
| ACCI, median (IQR) | 6 (5–7.75) | 6 (5–7) | 7 (6–9) | **0.007** | 6 (5–7) | 7 (6–8) | 0.101 |
| BMI[5], n (%) | | | | | | | |
| 18.5–24.9 (normal) | 41 (42.7) | 28 (48.3) | 13 (34.2) | 0.078 | 30 (42.9) | 11 (44.0) | 0.956 |
| 25–29.9 (overweight) | 38 (39.6) | 23 (39.7) | 15 (39.5) | | 28 (40.0) | 9 (36.0) | |
| ≥ 30 (obese) | 17 (17.7) | 7 (12.1) | 10 (26.3) | | 12 (17.1) | 5 (20.0) | |
| ASA class, n (%) | | | | | | | |
| ASA 1 | 2 (2.1) | 1 (1.7) | 1 (2.6) | **0.045** | 0 (0.0) | 1 (4.0) | **0.009** |
| ASA 2 | 28 (29.2) | 22 (37.9) | 6 (15.8) | | 27 (38.6) | 1 (4.0) | |
| ASA 3 | 56 (58.3) | 30 (51.7) | 26 (68.4) | | 37 (52.9) | 19 (76.0) | |
| ASA 4 | 10 (10.4) | 5 (8.6) | 5 (13.2) | | 6 (8.6) | 4 (16.0) | |
| Type of surgery, n (%) | | | | | | | |
| Ileocecal, right or left hemicolectomy | 38 (39.6) | 24 (41.4) | 14 (36.8) | 0.427 | 29 (41.4) | 8 (32.0) | 0.207 |
| Sigmoid, Hartmann | 21 (21.9) | 15 (25.9) | 6 (15.8) | | 17 (24.3) | 4 (16.0) | |
| Rectal | 34 (35.4) | 17 (29.3) | 17 (44.7) | | 21 (30.0) | 13 (52.0) | |
| Total colectomy or proctocolectomy | 3 (3.1) | 2 (3.4) | 1 (2.6) | | 3 (4.3) | 0 (0) | |
| Site of surgery, n (%) | | | | | | | |
| Colon or total | 61 (63.5) | 41 (70.7) | 20 (52.6) | 0.072 | 48 (68.6) | 12 (48.0) | 0.067 |
| Rectum | 35 (36.5) | 17 (29.3) | 18 (47.4) | | 22 (31.4) | 13 (52.0) | |
| Surgical approach, n (%) | | | | | | | |
| Laparoscopic | 71 (74.0) | 46 (79.3) | 25 (65.8) | 0.140 | 56 (80.0) | 14 (56.0) | **0.019** |
| Open or converted | 25 (26.0) | 12 (20.7) | 13 (34.2) | | 14 (20.0) | 11 (44.0) | |
| Stoma[6], n (%) | 33 (34.4) | 16 (27.6) | 17 (44.7) | 0.084 | 19 (27.1) | 14 (56.0) | **0.009** |
| Other surgical procedures, n (%) | 16 (16.7) | 8 (13.8) | 8 (21.1) | 0.351 | 9 (12.9) | 7 (28.0) | 0.118 |
| Cancer, n (%) | 72 (75.0) | 40 (69.0) | 32 (84.2) | 0.092 | 50 (71.4) | 21 (84.0) | 0.214 |
| Preoperative chemotherapy, n (%) | 11 (11.5) | 6 (10.3) | 5 (13.2) | 0.748 | 5 (7.1) | 6 (24.0) | **0.034** |

(*Continued*)

**Table 1.** (Continued)

| | All patients (n = 96) | CD < 2 POCs (n = 58) | CD ≥ 2 POCs (n = 38) | P-value | LOS ≤ P75[1] (n = 70) | LOS > P75[1] (n = 25) | P-value |
|---|---|---|---|---|---|---|---|
| Preoperative radiotherapy, n (%) | 13 (13.5) | 6 (10.3) | 7 (18.4) | 0.258 | 5 (7.1) | 8 (32.0) | **0.004** |

ACCI: Age Adjusted Charlson Comorbidity Index; ASA: American Society of Anesthesiologists; BMI: Body Mass Index; CCI: Charlson Comorbidity Index; CD: Clavien-Dindo severity grade; fTRST: Flemish version of the Triage Risk Screening Tool; IQR: interquartile range; LOS: (postoperative) length of stay; n: number; POCs: postoperative complications;

[1]75th percentile = 10.75 days (only patients discharged alive were considered);

[2]≥5 medications;

[3]hospitalised in the last 3 months;

[4]Nutrition Risk Screening according to the NRS-2002;

[5]none with BMI <18.5 = underweight;

[6]colo- or ileostomy

Dindo ≥2 postoperative complications. The need for urinary catheter (re)placement was low: 5/92 patients (5.4%). The overall (mean) adherence for the 21 measured interventions was 87.9% (SD ±10.8).

## Outcome data

The median postoperative LOS was 7 days (IQR 5–10.8). Sixty-one patients (63.5%) had in-hospital postoperative complications, of whom 38 (39.6%) Clavien-Dindo ≥2 and 13 (13.5%) Clavien-Dindo ≥3 (Table 3). The median Comprehensive Complication Index was 8.7 (IQR 0–24.2). Details on individual postoperative complications can be found in S2 Table. Classification into specific disease categories showed the following in-hospital complication rates: ileus or gastroparesis with nasogastric tube placement 13.5%, urinary tract infection 9.3%, urinary retention 8.3%, medically treated confusion 5.2%, cardiac arrhythmia 5.2%, heart failure treated with diuretics 3.1%, pneumonia 3.1%, catheter-related bloodstream infection 2.1%, myocardial infarction 1%. None of the patients had deep venous thrombosis or pulmonary embolism. At 30 days post-surgery two patients (2.1%) had died and 12 patients were readmitted (12.6%). Eight patients (8.3%) had unplanned reoperations within 30 days of the initial operation. Six of them had an anastomotic leak.

## Risk factors for postoperative complications and for prolonged hospital stay

All baseline characteristics (n = 23) were analyzed as risk factors in relation to the primary outcomes in univariable analysis (Table 1). Age, polypharmacy, ACCI, and ASA class were found to be significantly associated with Clavien-Dindo ≥2 complications. Polypharmacy was an independent risk factor in the forward logistic regression analysis (OR 3.4, 95% CI 1.3–8.9, p = 0.013), together with the site of surgery (OR 2.7, 95% CI 1.1–6.8, p = 0.032). Age, fTRST, impaired mobility, polypharmacy, ASA class, surgical approach, creation of a stoma, preoperative chemotherapy and preoperative radiotherapy were significantly associated with prolonged postoperative LOS (Table 1). Age (OR 1.1, 95% CI 1.0–1.2, p = 0.020), ASA class (OR 2.7, 95% CI 1.1–6.7, p = 0.031) and preoperative radiotherapy (OR 10.6, 95% CI 2.6–44.3, p = 0.001) remained as independent risk factors in forward logistic regression analysis.

**Table 2. Adherence to the ERP.**

| | ERP component | Assessment of adherence | Result n (%) | n patients |
|---|---|---|---|---|
| 1 | Preadmission counselling | ERP consultation | 83 (86.5) | 96 |
| 2 | Preoperative optimization | n.a. | n.a. | 96 |
| 3 | Preoperative bowel preparation | No mechanical bowel preparation in colonic surgery | 60 (98.4) | 61[1] |
| 4 | Preoperative fasting | n.a. | n.a. | 96 |
| | Carbohydrate loading | Yes | 67 (69.8) | 96 |
| 5 | Pre-anesthetic medication | No sedative or anxiolytic premedication | 88 (91.7) | 96 |
| 6 | Thromboprophylaxis | Yes | 95 (99.0) | 96 |
| 7 | Antimicrobial prophylaxis | Yes | 96 (100) | 96 |
| 8 | Standard anesthesia protocol | n.a. | | |
| 9 | PONV prophylaxis | In patients with PONV risk $\geq$ 20% | 89 (94.7) | 94[2] |
| 10 | Laparoscopy and modifications of surgical access | Planned laparoscopic operation | 81 (84.4) | 96 |
| 11 | Nasogastric tubes | No NGT after reversal of anesthesia | 92 (95.8) | 96 |
| 12 | Preventing intraoperative hypothermia | Yes | 84 (87.5) | 96 |
| 13 | Perioperative fluid management | Stop intravenous fluids $\leq$ POD 3 (provided removal of PCEA/PCIA)<br>• All patients<br>• *CD < 2 POCs** | 52 (54.7)<br><br>*37 (63.8)** | 95[3]<br><br>*58* |
| 14 | Drain peritoneal cavity | No abdominal drain in colonic surgery | 50 (82.0) | 61[1] |
| 15 | Urinary catheter | Removal urinary catheter per protocol: laparoscopic colon POD 1, open colon or rectum POD 3 (only after PCEA removal)<br>• All patients<br>• *CD < 2 POCs** | 69 (77.5)<br><br>*50 (87.7)** | 89[3,4,5]<br><br>*57* |
| 16 | Ileus prevention | n.a. | n.a. | 96 |
| 17 | Multimodal opioid sparing postop analgesia[8] | PCEA in patients without coagulation disorders for whom open surgery was planned | 8 (88.9) | 9[6] |
| 18 | Early nutrition | Oral food POD 1 | 79 (84.9) | 93[7] |
| | | Oral food POD 2 | 83 (89.2) | 93[3,7] |
| | | Oral food POD 3 | 82 (90.1) | 91[3,7] |
| 19 | Postoperative glycemic control | Glucose day profile in patients with diabetes | 19 (95.0) | 20[8] |
| 20 | Early mobilization | Out of bed POD 1 | 78 (83.9) | 93[7] |
| | | Out of bed POD 2 | 87 (93.6) | 93[3,7] |
| | | Out of bed POD 3 | 90 (98.9) | 91[3,7] |
| **Overall adherence, mean (± SD)** | | | **87.9% (± 10.8)** | |

CD: Clavien-Dindo severity grade; ERP: enhanced recovery program; n: number; n.a.: adherence was not assessed; NGT: nasogastric tube; PCEA: patient controlled epidural analgesia; PCIA: patient controlled intravenous analgesia; POCs: postoperative complications; POD: postoperative day; PONV: postoperative nausea and vomiting; SD: standard deviation; excluded from the analysis:

[1]rectal surgery (n = 35),

[2]missing data (n = 2),

[3]deceased patient (n = 1),

[4]no urinary catheter (n = 3),

[5]permanent catheter (n = 3),

[6]open surgery planned (n = 15) in patients with coagulation disorders (n = 6),

[7]ICU admission on the POD considered,

[8]no diabetes (n = 76); [8]all patients received paracetamol, restrictive use of NSAIDs due to age and comorbidities;

*this subgroup is not included in the calculation of the overall adherence

**Table 3. Primary and secondary outcomes.**

|  | n = 96 |
| --- | --- |
| Postoperative length of stay, median (IQR) | 7 (5–10.75) |
| In-hospital postoperative complications, n (%) | |
| • Clavien-Dindo grade 1 | 23 (24.0) |
| • Clavien-Dindo grade 2 | 25 (26.0) |
| • Clavien-Dindo grade 3 | 4 (4.2) |
| • Clavien-Dindo grade 4 | 8 (8.3) |
| • Clavien-Dindo grade 5 | 1 (1.0) |
| 30-day mortality rate, n (%) | 2 (2.1) |
| 30-day readmission rate[1], n (%) | 12 (12.6) |
| 30-day (unplanned) reoperation rate, n (%) | 8 (8.3) |

IQR: interquartile range; n: number;

[1] n = 95, only patients discharged alive were considered

## Discussion

This retrospective study analyzed ERP adherence, outcomes, and risk factors for postoperative complications and prolonged postoperative LOS in older patients undergoing colorectal surgery in an ERP. Despite a high overall adherence rate to the ERP in our study, postoperative adverse events in our older patient population were common. In-hospital postoperative complications graded as Clavien-Dindo $\geq 2$ were present in every two out of five patients. Site of surgery and polypharmacy were independent predictors of $\geq$ grade 2 postoperative complications, while age, ASA class and preoperative radiotherapy were independent predictors of LOS $>75^{\text{th}}$ percentile.

Compared with two multicenter studies in adult patients undergoing colorectal cancer resections, the mean adherence rate of 87.9% to the ERP in our study is considered as high. Van Zelm et al. observed adherence rates above 80% for only 8 of 43 considered interventions, while in the study by the ERAS Compliance Group the mean adherence to 13 interventions was 76.6% [26, 27]. The lowest adherence rates in our study were found for preoperative carbohydrate loading and for timely removal of intravenous lines or urinary catheters. A possible explanation for omitting the carbohydrate drink could be that surgeries may take place earlier than scheduled and logistic reasons might play a role. Having diabetes did not significantly affect carbohydrate loading. This is consistent with the ERAS® guideline that allows carbohydrate loading if given along with diabetes medication. Thirty-six percent of patients without postoperative complications needing treatment (Clavien-Dindo grades 0 and 1) and without PCEA or PCIA still had an intravenous line by POD 3. This could have been due to the presence of electrolyte disturbances, insufficient fluid intake or excessive fluid losses, but these factors were not considered during the data collection. Moreover, a short enquiry on the ward learned that nurses are reluctant to remove catheters and intravenous lines, despite being encouraged to do so, because they anticipate reinsertion later on.

Overall postoperative complication rates and median postoperative LOS were quite high. In a review performed by our team that summarized outcomes of ERPs in older patients ($\geq 65$ years) after colorectal surgery, the median overall morbidity was 23.5% and the median postoperative LOS was 6 days [12]. Conducting this review showed that it is challenging to compare postoperative complications among studies, because in- and exclusion criteria differ and because complications are not defined or assessed in a uniform way. Many studies only assess

a selection of complications or range complications by organ system. Post-hoc classification of postoperative complications in our study into specific disease categories (see Results section) showed acceptable rates of individual complications. Due to decreased physiologic reserves and impaired homeostasis in older patients, the occurrence of postoperative complications often leads to a cascade of events, for example pneumonia, successively leading to cardiac ischemia, heart failure, renal insufficiency, electrolyte disturbances, and ileus. In our study, complications induced by a prior adverse event or its treatment were considered as individual complications, as suggested by Clavien et al. in a qualitative study that was performed to clarify controversies about the application of the Comprehensive Complication Index [28]. We have not found any study reporting the Comprehensive Complication Index in older patients undergoing elective colorectal surgery to compare our results with. The median postoperative LOS in this study was 1 day longer than the median postoperative LOS in our review [12]. This might be explained by variations in in- and exclusion criteria between our study and some of the studies included in the review, e.g. some studies included colonic procedures only, laparoscopic procedures only, excluded patients with intensive care stay, with multi-organ resection. Proactively considering time to readiness for discharge (TRD) on a daily basis could trigger earlier discharge [29]. There was no formal registration of TRD at the time of this study in our electronic medical records, neither was there a registration of discharge destination. Prolonged LOS might also reflect outflow difficulties, such as limited institutional rehabilitation capacity and insufficient elderly care facilities in our region.

Preoperative risk prediction in older patients is another challenge. Our study identified site of surgery and polypharmacy as independent risk factors for postoperative complications. Higher age, ASA class and preoperative radiotherapy were independent risk factors of prolonged postoperative LOS. In line with data from an inpatient national database in the United States, increasing age was negatively associated with postoperative complications and prolonged postoperative LOS in univariable analysis [30]. Although age was not an independent risk factor for postoperative complications, this study does underline the fact that older persons need special attention in the perioperative period due to a high prevalence of multimorbidity (60.4%), polypharmacy (62.5%) and ASA $\geq$3 classifications (68.7%). The fTRST showed a trend to predict postoperative complications (p = 0.066) and was associated with prolonged postoperative LOS in univariable analysis (p = 0.006). A prospective study in emergency abdominal surgery observed an association between the fTRST (cut-off $\geq$2), postoperative complications and postoperative LOS [31].

The findings of this study should be interpreted within the context of its design. First, it concerns a single center study with a small sample, focusing on short-term clinical outcomes. The results should therefore be viewed as hypothesis-generating to conduct future studies. Second, all data were retrospectively collected from the electronic medical records and the adherence rates are therefore based on what has been registered. Strengths of this study are the detailed reporting of the applied ERP components and of all postoperative medical events that occurred. Another strength is the fact that ERP components were listed unambiguously and in accordance with the ERAS® guidelines, which facilitates comparison with future studies.

This study will serve as a baseline for a quality improvement project in our center, in which we will implement surgical-geriatric co-management in the abdominal surgery department, followed by a mixed-methods prospective observational study. Future studies should incorporate frailty screening and geriatric assessment into the patients' baseline assessment to be able to report their biological age in addition to chronological age and should incorporate these assessments in risk prediction models. At risk patients should be targeted with individually tailored geriatric interventions to prevent or manage adverse outcomes.

## Conclusions

This study shows that ERPs are feasible with good adherence in older persons undergoing colorectal surgery. ERPs should be implemented in this patient group without reservations. The advantage of optimizing ERP-implementation in the older patient group could be larger than in younger patients, given the non-negligible occurrence of adverse postoperative outcomes and of baseline risk factors for adverse outcomes in this patient population.

## Supporting information

**S1 Table. Standard ERP in the University Hospitals Leuven for colorectal surgical patients in 2017.**
(DOCX)

**S2 Table. Postoperative complications in detail.**
(DOCX)

**S1 Data.**
(XLSX)

## Acknowledgments

The authors would like to thank Camille Goossens for her contribution to the data acquisition, Ingrid Van Dessel for critical appraisal of the data and Annouschka Laenen for providing statistical advice.

## Author Contributions

**Conceptualization:** Katleen Fagard, Albert Wolthuis, Marleen Verhaegen, Johan Flamaing, Mieke Deschodt.

**Data curation:** Katleen Fagard.

**Formal analysis:** Katleen Fagard.

**Investigation:** Katleen Fagard.

**Methodology:** Katleen Fagard, Mieke Deschodt.

**Project administration:** Katleen Fagard.

**Supervision:** Albert Wolthuis, Johan Flamaing, Mieke Deschodt.

**Visualization:** Mieke Deschodt.

**Writing – original draft:** Katleen Fagard.

**Writing – review & editing:** Albert Wolthuis, Marleen Verhaegen, Johan Flamaing, Mieke Deschodt.

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
