## [Decision Letter · Decision Letter 0]

19 Feb 2020

PONE-D-20-01592

A retrospective observational study of enhanced recovery after surgery in older patients undergoing elective colorectal surgery.

PLOS ONE

Dear Dr. Fagard,

Thank you for submitting your manuscript to PLOS ONE. After careful consideration, we feel that it has merit but does not fully meet PLOS ONE’s publication criteria as it currently stands. Therefore, we invite you to submit a revised version of the manuscript that addresses the points raised during the review process.

We would appreciate receiving your revised manuscript by Apr 04 2020 11:59PM. To enhance the reproducibility of your results, we recommend that if applicable you deposit your laboratory protocols in protocols.io, where a protocol can be assigned its own identifier (DOI) such that it can be cited independently in the future. For instructions see: http://journals.plos.org/plosone/s/submission-guidelines#loc-laboratory-protocols

We look forward to receiving your revised manuscript.

Kind regards,

Yan Li

Academic Editor

PLOS ONE

Journal Requirements:

Reviewers' comments:

Reviewer's Responses to Questions

**Comments to the Author**

1. Is the manuscript technically sound, and do the data support the conclusions?

Reviewer #1: Yes

Reviewer #2: Yes

2. Has the statistical analysis been performed appropriately and rigorously? 

Reviewer #1: Yes

Reviewer #2: N/A

3. Have the authors made all data underlying the findings in their manuscript fully available?

Reviewer #1: Yes

Reviewer #2: Yes

4. Is the manuscript presented in an intelligible fashion and written in standard English?

Reviewer #1: Yes

Reviewer #2: Yes

5. Review Comments to the Author

Reviewer #1: Comments: This study retrospectively reported enhanced recovery programs (ERP) in elective colorectal surgery in older patients. It is a very interesting issue that was not answered in this population. 96 patients were included from a single center. ERP adherence in older patients is high and feasible. The postoperative complications and prolonged postoperative length of stay are acceptable.

Some questions are raised for adherence to 21 ERP interventions in this study:

2) no mechanical bowel preparation in colonic surgery (116 line): why mechanical bowel preparation not applied in these older patients? The incidence of anastomotic leak in this study is occurred in 6 out of 96 patients.

12) no abdominal drain in colonic surgery (123 line): If there are some remaining fluid or SSI in the abdomen/pelvis, how to handle these conditions without drains?

15) oral food on POD 1 (126 line): How many prophylactic stomas were created after colectomy? I wonder whether these 6 cases with anasto leak start oral food on POD 1.

Reviewer #2: Review for Plos One

ERAS in elderly patients over 70 YO

Authors make a retrospective single center observational study of patients over 70 years old, who underwent elective colorectal surgery following Enhanced Recovery Protocols (ERP).

The abstract is clear and well written. The abstract points to the existence of ERP, but also to its limitations for elderly patients. Therefore, the aims of the study link well to the background, and it is also well shown in the main manuscript.

The results of the abstract are also well briefly demonstrated, showing the good adherence of elderly patients to ERP, the complications with Clavien Dindo score and length of stay, which was long for an ERP. Finally, the multivariable analysis indicates the risk factors for complications and for prolonged length of stay.

The conclusions of the abstract are also clear and explain well the high adherence consequently the feasibility of ERP for elderly patients undergoing colorectal surgery.

The manuscript is also well written, with a comprehensive English.

The introduction of the manuscript is concise and goes directly to the point of ERP in elderly patients which is scarce in the literature and links well with the objectives of the study.

The materials and methods session is also well explained and divided. Figure one brings the flowchart of the study and exclusions which are also well pointed. However, figure 1 is not cited in the text, and I would consider citing in the sample subtitle after line 79.

Data analysis: seems ok to me, despite my experience in statistics is limited to the statistical methods applied. The methods look correct so far to me.

The results session is well fragmented to facilitate the understanding.

The conversion rates for were acceptable, but relatively high, specially for the rectal operations (16%). What were the criteria of conversion utilized in the study and how do the authors explain this conversion rates?

The adherence to ERP was quite high, congratulations. Although, is there any explanation for lower adherence rates for urinary catheter removal, carbohydrate loading and cassation of IV fluids?

I understand that patients age has an importance influence in postoperative outcomes and ERAS protocol applications, however in my understanding the mean LOS of 7 days seems to long for a fast track protocol despite the low complication rates. How do the authors explain these results? Also, analyzing the risk factors for prolonged LOS do the authors think they could represent an impairment on ERP and fast track protocols?

The discussion session is comprehensive and up-to-date. In line 241 why do authors cite (see results) instead of putting the reference of your previous publication? Also you try to answer the question I did before in the results about the LOS, however it was still not very clear for me? Specially after seeing the reference of your prior review.

I liked the criticism of the authors about your own production at the limitations of the study and when you say it “should be viewed as hypothesis-generating to conduct future studies”. Also, I liked the application this study will have for quality improvement in your center.

The conclusion of the manuscript is confused. I would follow the idea of the conclusion of the abstract. You don’t need to repeat some things you just wrote in the discussion, which are not conclusions.

Congratulations and good luck for the authors for this study and future ones.

6. PLOS authors have the option to publish the peer review history of their article (what does this mean?). If published, this will include your full peer review and any attached files.

Reviewer #1: Yes: Zixu Yuan

Reviewer #2: No

---

## [Author Response · Author response to Decision Letter 0]

28 Mar 2020

Reviewers' comments:

Reviewer #1

1. No mechanical bowel preparation in colonic surgery (116 line): why mechanical bowel preparation not applied in these older patients? The incidence of anastomotic leak in this study is occurred in 6 out of 96 patients.

Response: We understand the reviewer’s concern. The ERAS guidelines 20121 (which were the guidelines applied in our hospital at the time of our study) recommend to avoid mechanical bowel preparation (MPB) in colonic surgery because it has no clinical advantages and can cause dehydration, electrolyte disturbances, discomfort for the patient, prolonged ileus, and spillage of residual bowel contents into the abdominal space during the operation (evidence level: high, recommendation grade: strong). The latest ERAS guidelines (2018)2 repeat the same recommendation and refer to a recent meta-analysis comparing adult patients receiving MBP with those receiving no MBP: no significant difference was found between the two groups in the rates of surgical site infections, anastomotic leaks, intra-abdominal collections, mortality, reoperation, length of stay. To clarify that ERP guidelines contain several interventions in contradiction with past common surgical practices, the following sentence was added to the introduction: ‘ERPs include evidence-based changes in traditional care, such as pre-admission counselling, avoidance of mechanical bowel preparation, shortened fasting, a carbohydrate drink two hours before surgery, avoidance of fluid overload, minimal invasive surgery, avoidance or early removal of drains and catheters, opioid-sparing multimodal analgesia, early feeding and mobilization.’

1. Gustafsson et al. Guidelines for perioperative care in elective colonic surgery: Enhanced Recovery After Surgery (ERAS(R)) Society recommendations. Clinical nutrition (Edinburgh, Scotland). 2012;31(6):783-800.

2. Gustafsson et al. Guidelines for Perioperative Care in Elective Colorectal Surgery: Enhanced Recovery After Surgery (ERAS(R)) Society Recommendations: 2018. World journal of surgery. 2019;43(3):659-95.

2. No abdominal drain in colonic surgery (123 line): If there are some remaining fluid or SSI in the abdomen/pelvis, how to handle these conditions without drains?

Response: The ERAS guidelines 2012 discourage the use of drains because there is no proven benefit on clinical or radiological anastomotic dehiscence, wound infection, re-operation, extra-abdominal complications or mortality. Peri-anastomotic drainage does not allow early detection of anastomotic leak, nor does it help to control anastomotic dehiscence (evidence level: high, recommendation grade: strong). The latest ERAS guidelines (2018) repeat the same recommendation and cite more recent studies to underline the recommendation. In our experience spontaneous resorption of remaining fluids occurs. Moreover, avoidance or early removal of catheters and drains facilitates early mobilisation, which is also part of the ERAS protocol. 

3. Oral food on POD 1 (126 line): How many prophylactic stomas were created after colectomy? I wonder whether these 6 cases with anastomotic leak start oral food on POD 1.

Response: In our center no prophylactic stomas are created after elective colectomy. Five of six patients with anastomotic leak started oral food on POD 1. Their anastomotic leaks occurred on postoperative day 3, 9, 10, 12 and 17. The sixth patient was not eating on POD1 and had a reoperation for an anastomotic leak on POD 2. The rates of anastomotic leaks in our study (6.25%) might seem high. In studies with comparable in- and exclusion criteria we have found percentages between 0 and 9%.1-5 In some studies it was not clear if the complication was registered upon discharge or during follow-up. In our study in four patients an anastomotic leak occurred during hospitalisation, and two patients were readmitted for anastomotic leaks after discharge. The ERAS 2012 guidelines state that early oral feeding (versus nil by mouth) reduces the risk of infection and length of stay, and is not associated with an increased risk of anastomotic dehiscence. The risk of vomiting might increase, but therefore multi-modal anti-ileus and nausea and vomiting prophylaxis is recommended and part of the ERAS protocol. The ERAS 2018 guideline refers to a study that confirmed the safety of an early oral diet (4h after surgery) in patients with a new non-diverted colorectal anastomosis. 

1. Gonzalez-Ayora et al. Enhanced recovery care after colorectal surgery 

in elderly patients. Compliance and outcomes of a multicenter study from the

Spanish working group on ERAS. Int J Colorectal Dis. 2016 Sep;31(9):1625-31. 

2. Braga et al. PeriOperative Italian Society Group. Enhanced recovery pathway in elderly

patients undergoing colorectal surgery: is there an effect of increasing ages?

Results from the perioperative Italian Society Registry. Updates Surg. 2018

Mar;70(1):7-13. 

3. Tejedor et al. Short-term outcomes and benefits of ERAS program in elderly

patients undergoing colorectal surgery: a case-matched study compared to

conventional care. Int J Colorectal Dis. 2018 Sep;33(9):1251-1258. 

4. Ostermann et al. Randomized Controlled Trial of Enhanced Recovery Program Dedicated to

Elderly Patients After Colorectal Surgery. Dis Colon Rectum. 2019

Sep;62(9):1105-1116. 

5. Joris et al. Elderly patients over 70 years

benefit from enhanced recovery programme after colorectal surgery as much as

younger patients. J Visc Surg. 2020 Feb;157(1):23-31. 

Reviewer #2

1. Figure one brings the flowchart of the study and exclusions which are also well pointed. However, figure 1 is not cited in the text, and I would consider citing in the sample subtitle after line 79.

Response: Figure 1 is referred to in the text: not in the materials and methods section but in the results section ‘description of the sample’ (figure 1 reports the results of the study recruitment).

2. The conversion rates were acceptable, but relatively high, especially for the rectal operations (16%). What were the criteria of conversion utilized in the study and how do the authors explain this conversion rates?

Response: Although the ERAS guidelines state that conversion rates (in colorectal cancer surgery) of less than 10% should be achievable, we do believe that the reported conversion rates in our study are acceptable. In a large database analysis (National Inpatient Sample database, United states)1 that included 337 732 laparoscopic colorectal resections (2009-2012), 48 265 procedures (14.3%) were converted to open surgery and a conversion of 31.2% was observed for proctectomy. In another analysis of the same database (2009-2010) by the same group2 Crohn's disease, prior abdominal surgery, proctectomy, malignant pathology, emergent surgery, obesity, and ulcerative colitis showed the highest conversion rates. In our study 10.0% of colonic operations and 16.1% of rectal operations were converted from laparoscopic to open surgery. The reasons for conversion were added in the results section: ‘The reasons for conversion were obesity (n=2), extensive adhesions (n=5), extensive malignancy (n=2), and difficult splenic flexure mobilisation (n=1).’ 

1. Moghadamyeghaneh et al. Outcomes of conversion of laparoscopic colorectal surgery to open surgery. JSLS. 2014;18(4):e2014.

2. Masoomi et al. Risk Factors for Conversion of Laparoscopic Colorectal Surgery to Open Surgery: Does Conversion Worsen Outcome? World J Surg 39, 1240–1247 (2015).

3. The adherence to ERP was quite high, congratulations. Although, is there any explanation for lower adherence rates for urinary catheter removal, carbohydrate loading and cassation of IV fluids?

Response: Thank you for this remark. We agree to provide further explanation in the discussion. The second paragraph of the discussion has been rewritten as follows (new parts in italics): ‘… The lowest adherence rates in our study were found for preoperative carbohydrate loading and for timely removal of intravenous lines and urinary catheters. A possible explanation for omitting the carbohydrate drink could be that surgeries may take place earlier than scheduled and logistic reasons might play a role. Having diabetes did not significantly affect carbohydrate loading. This is consistent with the ERAS® guideline that allows carbohydrate loading if given along with diabetes medication. Thirty-six percent of patients without postoperative complications needing treatment (Clavien-Dindo grades 0 and 1) and without PCEA or PCIA still had an intravenous line by POD 3. This could have been due to the presence of electrolyte disturbances, insufficient fluid intake or excessive fluid losses, but these factors were not considered during the data collection. Moreover, a short enquiry on the ward learned that nurses are reluctant to remove catheters and intravenous lines, despite being encouraged to do so, because they anticipate reinsertion later on.’

4. I understand that patients age has an importance influence in postoperative outcomes and ERAS protocol applications, however in my understanding the mean LOS of 7 days seems to long for a fast track protocol despite the low complication rates. How do the authors explain these results? In the discussion you try to answer the question I did before in the results about the LOS, however it was still not very clear for me? Specially after seeing the reference of your prior review.

Response: We appreciate this comment. The median postoperative LOS was 7 days in this study, compared to 6 days in the review.1 Apart from the explanations provided in the text, there may be some additional explanations. The patients included in this study (patients ≥ 70y) might have been somewhat older than the patients included in the review, which included patients ≥ 65y. Moreover, some of the included articles in the review had stricter inclusion criteria than this study, which might also explain a shorter median length of stay (e.g. colonic surgery only, laparoscopic surgery only, exclusion of patients with metastatic cancer / intensive care admission / multi-organ resection / cognitive impairment / ASA 4). These are assumptions that cannot be proven because the studies included in the review do not provide sufficient information to allow pooling and subanalyses of the results. We have added the following sentence to the discussion: ‘This might be explained by variations in in- and exclusion criteria between our study and some of the studies included in the review, e.g. some studies included colonic procedures only, laparoscopic procedures only, excluded patients with intensive care stay, with multi-organ resection. Prolonged LOS might also be due to outflow difficulties …’ 

1. Fagard et al. A systematic review of the intervention components, adherence and outcomes of enhanced recovery programmes in older patients undergoing elective colorectal surgery. BMC geriatrics. 2019;19(1):157.

5. Analyzing the risk factors for prolonged LOS do the authors think they could represent an impairment on ERP and fast track protocols? 

Response: This question was not entirely clear to us, but we suppose the reviewer wants to know if reduced ERP adherence at individual patient level was a risk factor for prolonged LOS in our study. The response to this question is that we studied overall adherence. It was not our aim to study individual adherence per patient in relation our primary outcomes (complications ≥ grade 2 and prolonged LOS). We have found a review published in 20171 that tried to answer the question: They found that the most frequently identified predictor of prolonged LOS was reduced individual compliance with the ERAS protocol, both globally and with specific components such as delayed mobilisation and delayed resumption of oral intake. It is noteworthy that lower adherence in the postoperative period might reflect the development of complications. Therefore we think that lower adherence rates for early mobilisation and early feeding should not automatically be considered as an implementation failure, but could be due to unforeseen events. 

2. Messenger et al. Factors predicting outcome from enhanced recovery programmes in laparoscopic colorectal surgery: a systematic review. Surg Endosc 31, 2050–2071 (2017). 

6. The discussion session is comprehensive and up-to-date. In line 241 why do authors cite (see results) instead of putting the reference of your previous publication? 

Response: We apologize for the confusion: ‘see results’ refers to the results section of the current manuscript and does not refer to our previous publication. This was clarified in the text: ‘Post-hoc classification of postoperative complications in our study into specific disease categories (see results section) showed acceptable rates of individual complications.’

7. The conclusion of the manuscript is confused. I would follow the idea of the conclusion of the abstract. You don’t need to repeat some things you just wrote in the discussion, which are not conclusions. 

Response: We thank the reviewer for this constructive comment. The reviewer is right that the conclusion was broader than the findings of the study. To correct this, and in order to avoid repetition, the last two sentences of the conclusion have been removed.

---

## [Editor Report · Decision Letter 1]

23 Apr 2020

A retrospective observational study of enhanced recovery after surgery in older patients undergoing elective colorectal surgery.

PONE-D-20-01592R1

Dear Dr. Fagard,

We are pleased to inform you that your manuscript has been judged scientifically suitable for publication and will be formally accepted for publication once it complies with all outstanding technical requirements.

With kind regards,

Yan Li

Academic Editor

PLOS ONE
---

## [Editor Report · Acceptance letter]

29 Apr 2020

PONE-D-20-01592R1 

A retrospective observational study of enhanced recovery after surgery in older patients undergoing elective colorectal surgery. 

Dear Dr. Fagard:

I am pleased to inform you that your manuscript has been deemed suitable for publication in PLOS ONE. Congratulations! Your manuscript is now with our production department. 

With kind regards,

on behalf of

Dr. Yan Li 

Academic Editor

PLOS ONE